# Antiphospholipid Antibodies and Heart Failure with Preserved Ejection Fraction. The Multicenter ATHERO-APS Study

**DOI:** 10.3390/jcm10143180

**Published:** 2021-07-19

**Authors:** Daniele Pastori, Paul R. J. Ames, Massimo Triggiani, Antonio Ciampa, Vittoria Cammisotto, Roberto Carnevale, Pasquale Pignatelli, Tommaso Bucci

**Affiliations:** 1Department of Clinical, Internal, Anesthesiological and Cardiovascular Sciences, Sapienza University of Rome, 00155 Rome, Italy; vittoria.cammisotto@uniroma1.it (V.C.); pasquale.pignatelli@uniroma1.it (P.P.); 2Immune Response and Vascular Disease Unit, Nova University, 1099-085 Lisbon, Portugal; paxmes@aol.com; 3Dumfries and Galloway Royal Infirmary, Dumfries DG2 8RX, UK; 4Department of Internal Medicine, Division of Allergy and Clinical Immunology, University of Salerno, 84084 Salerno, Italy; mtriggiani@unisa.it; 5Centro Emostasi A.O.R.N. “SG Moscati”, 83100 Avellino, Italy; ciampa@inopera.it; 6Department of General Surgery and Surgical Specialties “Paride Stefanini”, Sapienza University of Rome, 00155 Rome, Italy; tommaso.bucci@uniroma1.it; 7Department of Medical-Surgical Sciences and Biotechnologies, Sapienza University of Rome, 04100 Latina, Italy; roberto.carnevale@uniroma1.it; 8Mediterranea Cardiocentro, 80133 Napoli, Italy

**Keywords:** antiphospholipid syndrome, HFpEF, echocardiography, heart failure

## Abstract

Background. The prevalence of heart failure with preserved ejection fraction (HFpEF) in patients with antiphospholipid syndrome (APS) is unknown. Methods. A prospective multicenter cohort study including 125 patients was conducted: 91 primary APS (PAPS), 18 APS-SLE, and 16 carriers. HFpEF was diagnosed according to the 2019 European Society of Cardiology criteria: patients with ≥5 points among major and minor functional and morphological criteria including NT-ProBNP > 220 pg/mL, left atrial (LA) enlargement, increased left ventricular filling pressure. Results. Overall, 18 (14.4%) patients were diagnosed with HFpEF; this prevalence increased from 6.3% in carriers to 13.2% in PAPS and 27.8% in APS-SLE. Patients with HFpEF were older and with a higher prevalence of hypertension and previous arterial events. At logistic regression analysis, age, arterial hypertension, anticardiolipin antibodies IgG > 40 GPL (odds ratio (OR) 3.43, 95% confidence interval (CI) 1.09–10.77, *p* = 0.035), anti β-2-glycoprotein-I IgG > 40 GPL (OR 5.28, 1.53–18.27, *p* = 0.009), lupus anticoagulants DRVVT > 1.25 (OR 5.20, 95% CI 1.10–24.68, *p* = 0.038), and triple positivity (OR 3.56, 95% CI 1.11–11.47, *p* = 0.033) were associated with HFpEF after adjustment for age and sex. By multivariate analysis, hypertension (OR 19.49, 95% CI 2.21–171.94, *p* = 0.008), age (OR 1.07, 95% CI 1.00–1.14, *p* = 0.044), and aβ2GPI IgG > 40 GPL (OR 8.62, 95% CI 1.23–60.44, *p* = 0.030) were associated with HFpEF. Conclusion. HFpEF is detectable in a relevant proportion of APS patients. The role of aPL in the pathogenesis and prognosis of HFpEF needs further investigation.

## 1. Introduction

Antiphospholipid syndrome (APS) is an autoimmune disorder including a wide range of conditions ranging from seronegative APS to primary APS (PAPS) and secondary systemic lupus erythematosus-associated APS (APS-SLE) [1]. APS is characterized by a significant morbidity and mortality, which are not prevented by current treatments [2,3]. Thus, an increased risk of venous and arterial vascular thrombotic events has been described despite oral anticoagulant treatment [4,5,6].

In addition to thrombotic complications, patients with APS may also have cardiac abnormalities, including systolic and diastolic dysfunction, which may be present both in patients with PAPS and APS-SLE [7,8]. In accordance, previous data suggested that heart failure (HF) may be one of the clinical manifestation of APS, but this evidence relies on small case series [9].

The N-terminal ProBNP (NT-ProBNP) is widely used for the diagnosis and monitoring of HF, and its plasma levels bear prognostic value [10]. An NT-ProBNP-guided patient management was associated with a lower incidence of cardiovascular events compared to standard of care [11].

The combined use of NT-ProBNP and echocardiography data allows a better characterization of HF phenotypes [12]. In particular, a previously unrecognized group of patients is represented by the so-called HF with preserved ejection fraction (HFpEF). These patients have a preserved left ventricular (LV) ejection fraction (EF) > 50%, with an increased level of NT-ProBNP and echocardiography evidence of structural heart disease, including left atrial (LA) enlargement and increased LV filling pressure [12]. The natural history of patients with HFpEF is complicated by a remarkably high incidence of cardiovascular complications [13]. Thus, the rate of total hospitalizations for heart failure and death from cardiovascular causes in the PARAGON-HF trial ranged from 12.8 to 14.6 per 100 patient-years [14].

Levels of NT-ProBNP as well as prevalence of HFpEF in patients with APS have never been reported.

To this aim, we investigated the presence of HFpEF in a cohort of consecutive patients with APS.

## 2. Materials and Methods

We derived our data from a multicenter ongoing prospective cohort study including consecutive APS patients from four centers: (1) Atherothrombosis Center of Department of Clinical Internal, Anesthesiologic and Cardiovascular Sciences of Sapienza University of Rome, (2) Fondazione APS—Anticorpi Antifosfolipidi ONLUS, Naples, (3) Centro Emostasi A.O.R.N. “SG Moscati”, Avellino, and (4) Division of Allergy and Clinical Immunology, University of Salerno, all in Italy. We included patients with confirmed diagnosis of APS by anticardiolipin (aCL), anti β-2-glycoprotein-I (aβ2GPI), or lupus anticoagulant (LAC) [15]. We also included a control group of patents with antiphospholipid antibody (aPL) but without thrombotic events (carriers group).

Exclusion criteria were active cancer or history/treatment with cardiotoxic drugs, history of myocardial infarction, or heart failure with reduced ejection fraction (HFrEF).

The diagnosis of diabetes [16], arterial hypertension [17], previous myocardial infarction [18], and HFrEF [12] were made according to the current international guidelines.

## 3. NT-ProBNP Measurement

NT-ProBNP levels were detected at baseline in all patients using a sandwich ELISA technology kit (FineTest^®^). Values were expressed as pg/mL. Intra- and inter-assay coefficients of variation were <8% and <10%, respectively.

## 4. Transthoracic Echocardiography

All resting transthoracic echocardiography exams were performed at baseline in left lateral decubitus with sequential analysis of the parasternal, apical, suprasternal, and subxiphoid windows. Echocardiographic parameters were assessed in conformity with the recommendations of the American Society of Echocardiography (ASE) [19]. The same two operators performed echocardiography examinations in all centers (DP, TB).

We collected the following echocardiographic parameter: diastolic interventricular septum (IVS) and posterior wall, LV diastolic diameter volume indexed to body surface area (BSA), relative wall thickness (RWT), LV mass (LVM)/BSA, LV ejection fraction (Simpson’s method), right ventricle diameter, LA diameter, LA area, LA volume/BSA, pulsed Doppler analysis of mitral flow (E wave, A wave, E/A ratio), lateral and septal mitral annular tissue Doppler (e’septal, e´lateral), and E/e´ratio. According to LVM/BSA and RWT, we classified LV geometry as (1) normal LV geometry, (2) concentric remodeling, (3) concentric hypertrophy, (4) eccentric hypertrophy.

The presence of HFpEF was diagnosed according to the 2019 consensus recommendation from the Heart Failure Association (HFA) of the European Society of Cardiology (ESC). Thus, patients with ≥5 points as the sum of major and minor criteria were diagnosed with HFpEF [20].

## 5. Ethical Statement

The study was conducted according to the principles embodied in the Declaration of Helsinki and was approved by local ethical board of Sapienza University of Rome (Ref: 4417, 2 March 2017). All patients provided written informed consent.

## 6. Statistical Analysis

Categorical variables were reported as counts (percentage); continuous variables were expressed as mean ± standard deviation. Independence of categorical variables was tested with the χ^2^ test. Student’s unpaired *t* tests and ANOVA tests were used to compare means. A first descriptive analysis of clinical and echocardiography characteristics according to aPL status, such as PAPS, APS-SLE, and carriers was performed. Then, patients were divided in two groups according to the presence or not of HFpEF. Univariable and multivariable logistic regression analysis were used to calculate the relative odds ratio (OR) and 95% confidence interval (95% CI) for each factor associated with the diagnosis of HFpEF. Only variables associated with HFpEF after adjustment for age and sex were inserted in the multivariable logistic regression analysis model. Only *p* values < 0.05 were considered as statistically significant. Analysis was performed using SPSS-25.0, SPSS Inc.

## 7. Results

The study included 130 consecutive patients with aPL; of these, 4 were excluded as presenting with HFrEF and 1 for missing data. Thus, the final cohort was composed of 125 patients: 91 primary APS, 18 APS-SLE, and 16 carriers. Previous arterial events were recorded in 39 patients: 30 stroke and 9 peripheral artery thrombosis. Previous VTE were recorded in 74 patients: 70 deep vein thrombosis/pulmonary embolism, 2 splanchnic vein thrombosis, and 2 cerebral vein thrombosis. Characteristics of patients are listed in Table 1.

Patients with APS-SLE were more frequently women, with a higher prevalence of hypertension and diabetes compared to the other groups (Table 1). Previous VTE was more frequent in PAPS patients, while APS-SLE patients had higher prevalence of arterial ischemic events (Table 1).

### Characteristics of Patients with HFpEF

In the whole cohort, 18 (14.4%) patients had HFpEF. Patients with HFpEF were more frequently affected by APS-SLE, older with a higher prevalence of previous arterial events (Table 2). Regarding auto-antibodies, a higher prevalence of aCL IgG > 4 0 GPL (64.7% vs. 37.7%, *p* = 0.036), aβ2GPI IgM > 40 MPL (29.4% vs. 10.5%, *p* = 0.032), and IgG > 40 GPL (64.7% vs. 37.1%, *p* = 0.032), as well as a LAC DRVVT > 1.25 (88.2% vs. 61.3%, *p* = 0.031) was present in patients with HFpEF compared to those without, respectively (Table 2). A trend towards a higher prevalence of triple positivity in HFpEF patients was found.

By logistic regression analysis (Table 3), age, hypertension, aCL IgG > 40 GPL (OR 3.43, 95% CI 1.09–10.77, *p* = 0.035), aβ2GPI IgG > 40 GPL (OR 5.28, 1.53–18.27, *p* = 0.009), LAC DRVVT > 1.25 (OR 5.20, 95% CI 1.10–24.68, *p* = 0.038), and triple positivity (OR 3.56, 95% CI 1.11–11.47, *p* = 0.033) were associated with HFpEF after adjustment for age and sex.

To better understand the relationship between the variables significantly associated with HFpEF, we performed a multivariate logistic regression analysis (Table 4). In this model, hypertension (OR 19.49, 95% CI 2.21–171.94, *p* = 0.008), age (OR 1.07, 95% CI 1.00–1.14, *p* = 0.044), and aβ2GPI IgG > 40 GPL (OR 8.62, 95% CI 1.23–60.44, *p* = 0.030) were associated with HFpEF.

## 8. Discussion

This is the first study investigating the prevalence and correlates of HFpEF in APS. Our study shows a clinically relevant prevalence of HFpEF in patients with APS; thus, in a middle-aged cohort of patients, the prevalence of HFpEF increased from 6.3% in carriers to 13.2% in PAPS and to 27.8% in APS-SLE.

Patients with HFpEF were older than those without, and the majority suffered arterial hypertension. The association between hypertension and HFpEF has been previously recognized, given that hypertension is the main determinant of increased LV filling pressure and diastolic dysfunction, which is a major criteria for the diagnosis of HFpEF [21]. Furthermore, the intense management of blood pressure was shown to reduce the incidence of HFpEF [22].

Regarding auto-antibodies profile, we found a higher proportion of increased aCL IgG, aβ2GPI IgG, and LAC positivity in the group of patients with HFpEF. This association is novel and suggest that APS patients may have an early cardiac involvement represented by the HFpEF. Our results provide new insight into the association between aPL and cardiovascular disease. A previous metanalysis showed an increased risk of recurrent events in patients with MI and antiphospholipid antibodies [23], and a recent work reported a prevalence of APS in 15.5% of patients with myocardial infarction and non-obstructive coronary arteries [24]. The association between aPL and HFpEF suggests that aPL may contribute to the microvascular endothelial dysfunction that characterizes the pathogenesis of HFpEF [21]. Of interest, age, IgG aβ2GPI > 40 GPL, and hypertension were independently associated with HFpEF. Elevated IgG aβ2GPI directly relates to endothelin-1 [25] and to isoprostane [26], two powerful vasoactive agents, and inversely relates to nitric oxide metabolites [27], leading to increased vasomotor tone and arterial hypertension. The release of reactive oxygen species mediated by aPL is responsible for increased oxidative stress [28], lipid peroxidation [29], loss of the biological activity of nitric oxide [30], and aPL modifications [31], all factors that contribute to the endothelial dysfunction status that characterize hypertension [32] and HFpEF [33]. The potential role of oxidative stress in HFpEF is also indirectly suggested by interventional studies with antioxidant compounds, showing an improvement in diastolic [34] and endothelial function [35,36].

In addition, a higher proportion of triple positivity was found in HFpEF patients, indicating increased potential thrombogenicity [37] in patients with HFpEF, as triple positive patients have been shown to have an increased risk of thrombotic events compared to those with single or double aPL positivity [38].

Remarkably, also 6.3% of aPL carriers were diagnosed with HFpEF. The potential usefulness of preventive therapeutic strategy with aspirin or anti-hypertensive drugs in this subgroup of subjects warrants further investigation.

Indeed, implications of our findings are that patients with aβ2GPI IgG > 40 and triple positivity should undergo a strict cardiology follow-up to early detect the onset of HFpEF, as this subgroup of patients may have a worse prognosis.

The limitations of this study include the observational design and the relatively small sample size that prevents us from drawing definite conclusion. Therefore, our results are to be regarded as hypothesis generating. Furthermore, there are several clinical issues that need to be addressed, such as the prognostic role of HFpEF in PAPS and APS-SLE patients, as well as the role of different aPL types in patients diagnosed with HFpEF.

However, the identification of HFpEF may help to refine the heterogenous clinical phenotypes of APS patients [39,40], potentially identifying those at higher risk of cardiovascular events.

In conclusion, a considerable proportion of patients with APS may have HFpEF. Long term studies to investigate the impact of HFpEF on cardiovascular outcomes in these patients are needed.

## Figures and Tables

**Table 1 jcm-10-03180-t001:** Characteristics of patients according to aPL status.

	PAPS (*n* = 91)	APS-SLE (*n* = 18)	Carriers (*n* = 16)	*p* among Groups
**Age (years)**	51.4 ± 14.0	52.2 ± 14.8	47.2 ± 14.1	0.501 ^#^
**Women (%)**	60 (65.9)	16 (88.9)	13 (81.3)	0.092 ^§^
**Hypertension (%)**	42 (46.2)	14 (77.8)	4 (25.0)	0.007 ^§^
**Diabetes (%)**	3 (3.3)	3 (16.7)	0 (0.0)	0.033 ^§^
**Smoking (%)**	21 (23.1)	2 (11.1)	1 (6.3)	0.185 ^§^
**Previous arterial events (%)**	31 (34.1)	8 (44.4)	0 (0.0)	0.011 ^§^
**Previous VTE (%)**	65 (71.4)	9 (50.0)	0 (0.0)	<0.001 ^§^
**HFpEF (%)**	12 (13.2)	5 (27.8)	1 (6.3)	0.167 ^§^
**NT-ProBNP (pg/mL)**	455.9 ± 118.6 *	537.1 ± 116.0 **	288.2 ± 41.0	<0.001 ^#^
**Treatments**				
**Hydroxychloroquine (%)**	13 (14.3)	8 (44.4)	5 (31.3)	0.009 ^§^
**Proton pump inhibitors (%)**	23 (25.3)	11 (61.1)	4 (25.0)	0.009 ^§^
**Corticosteroids (%)**	12 (13.2)	14 (77.8)	5 (31.3)	<0.001 ^§^
**Antiplatelet drugs (%)**	18 (19.8)	3 (16.7)	8 (50.0)	0.024 ^§^
**Oral anticoagulants (%)**	60 (65.9)	10 (55.6)	0 (0.0)	<0.001 ^§^
**Statins (%)**	20 (22.0)	0 (0.0)	2 (12.5)	0.069 ^§^
**ACEi/ARBs (%)**	33 (36.3)	10 (55.6)	3 (18.8)	0.083 ^§^
**Beta blockers (%)**	21 (23.1)	6 (33.3)	2 (12.5)	0.356 ^§^
**Calcium channel blockers (%)**	8 (8.9)	6 (33.3)	0 (0.0)	0.004 ^§^
**Diuretics (%)**	17 (18.7)	2 (11.1)	2 (12.5)	0.651 ^§^
**Autoantibodies**				
**aCL IgG > 40 GPL (%)**	41 (46.1)	6 (33.3)	4 (25.0)	0.217 ^§^
**aCL IgM > 40 MPL (%)**	15 (16.9)	3 (16.7)	4 (25.0)	0.728 ^§^
**aβ2GPI IgG > 40 GPL (%)**	36 (40.9)	8 (44.4)	6 (37.5)	0.919 ^§^
**aβ2GPI IgM > 40 MPL (%)**	11 (12.5)	2 (11.1)	3 (18.8)	0.764 ^§^
**LAC DRVVT > 1.25 (%)**	66 (74.2)	12 (66.7)	2 (12.5)	<0.001 ^§^
**Triple positivity (%)**	41 (46.1)	6 (33.3)	1 (6.3)	0.009 ^§^
**Echocardiography measurements**				
**Ejection fraction (%)**	60.1 ± 7.2	56.9 ± 6.3	61.7 ± 6.1	0.122 ^#^
**LVED volume/BSA (mL/m^2^)**	53.1 ± 13.2	54.7 ± 9.8	52.7 ± 9.7	0.881 ^#^
**LVM/BSA (g/m^2^)**	81.2 ± 22.9	80.7 ± 20.3	77.7 ± 28.5	0.861 ^#^
**Normal LV geometry (%)**	54.4	55.6	68.8	0.929 ^§^
**Concentric remodeling (%)**	25.6	22.2	18.8	
**Concentric hypertrophy (%)**	8.9	5.6	6.3	
**Eccentric hypertrophy (%)**	11.1	16.7	6.3	
**LA diameter (mm)**	35.9 ± 5.9	38.8 ± 6.6	32.7 ± 4.4	0.012 ^#^
**LA area (cm^2^)**	18.9 ± 4.3	20.3 ± 6.2	17.2 ± 3.5	0.166 ^#^
**LA volume/BSA (mL/m^2^)**	28.7 ± 8.6	32.7 ± 12.8	22.9 ± 7.4	0.010 ^#^
**E/A ratio**	1.22 ± 0.49	1.04 ± 0.32	1.25 ± 0.48	0.359 ^#^
**e’ septal**	0.09 ± 0.03	0.09 ± 0.02	0.10 ± 0.02	0.543 ^#^
**e’ lateral**	0.11 ± 0.04	0.11 ± 0.04	0.11 ± 0.03	0.947 ^#^
**Mean E/e’ ratio**	7.80 ± 2.49	9.09 ± 3.27	7.35 ± 1.27	0.103 ^#^

^#^ ANOVA test; ^§^ chi squared test; * *p* = 0.006 vs. carriers; ** *p* < 0.001 vs. carriers. ACEi: angiotensin-converting enzyme inhibitors; APS: antiphospholipid syndrome; ARBs: angiotensin receptor blockers; aβ2GPI: anti beta-2-glycoprotein-I antibody; aCL: anticardiolipin antibody; BSA: body surface area; DRVVT: dilute Russel’s viper venom time; HFpEF: heart failure with preserved ejection fraction; LA: left atrium; LAC: lupus anticoagulant; LVED: left ventricular end-diastolic; LVM: left ventricular mass; PAPS: primary APS; SLE: systemic lupus erythematosus; VTE: venous thromboembolism.

**Table 2 jcm-10-03180-t002:** Characteristics of patients according to the presence of heart failure with preserved ejection fraction (HFpEF).

	HFpEF No (*n* = 107)	HFpEF Yes (*n* = 18)	*p* among Groups
**PAPS (%)**	79 (73.8)	12 (66.7)	0.167 ^§^
**APS-SLE (%)**	13 (12.1)	5 (27.8)	
**Carriers (%)**	15 (14.0)	1 (5.6)	
**Age (years)**	49.8 ± 14.3	58.3 ± 10.1	0.017 ^#^
**Women (%)**	79 (73.8)	10 (55.6)	0.113 ^§^
**Hypertension (%)**	43 (40.2)	17 (94.4)	<0.001
**Diabetes (%)**	4 (3.7)	2 (11.1)	0.176 ^§^
**Smoking (%)**	21 (19.6)	3 (16.7)	0.768 ^§^
**Previous arterial events (%)**	29 (27.1)	10 (55.6)	0.016 ^§^
**Previous VTE (%)**	67 (62.6)	7 (38.9)	0.058 ^§^
**Treatments**			
**Hydroxychloroquine (%)**	19 (17.8)	7 (38.9)	0.041 ^§^
**Proton pump inhibitors (%)**	31 (29.0)	7 (38.9)	0.397 ^§^
**Corticosteroids (%)**	24 (22.4)	7 (38.9)	0.135 ^§^
**Antiplatelet drugs (%)**	23 (21.5)	6 (33.3)	0.271 ^§^
**Oral anticoagulants (%)**	61 (57.0)	9 (50.0)	0.579 ^§^
**Statins (%)**	17 (15.9)	5 (27.8)	0.220 ^§^
**ACEi/ARBs** **(%)**	34 (31.5)	12 (66.7)	0.005 ^§^
**Beta blockers (%)**	17 (15.9)	12 (66.7)	<0.001 ^§^
**Calcium channel antagonists (%)**	9 (8.5)	5 (27.8)	0.017 ^§^
**Diuretics (%)**	14 (13.1)	7 (38.9)	0.007 ^§^
**Autoantibodies**			
**aCL IgG > 40 GPL (%) ***	40 (37.7)	11 (64.7)	0.036 ^§^
**aCL IgM > 40 MPL (%) ***	18 (17.0)	4 (23.5)	0.513 ^§^
**aβ2GPI IgG > 40 GPL (%) ***	39 (37.1)	11 (64.7)	0.032 ^§^
**aβ2GPI IgM > 40 MPL (%) ***	11 (10.5)	5 (29.4)	0.032 ^§^
**LAC DRVVT > 1.25 (%) ***	65 (61.3)	15 (88.2)	0.031 ^§^
**Triple positivity (%) ***	38 (35.8)	10 (58.8)	0.071 ^§^
**Echocardiography measurements**			
**Ejection fraction (%)**	60.7 ± 6.4	55.1 ± 8.6	0.001 ^#^
**LVED volume/BSA (mL/m^2^)**	52.1 ± 12.1	60.1 ± 11.7	0.011 ^#^
**LVM/BSA (g/m^2^)**	75.7 ± 18.8	109.8 ± 26.1	<0.001 ^#^
**Normal LV geometry (%)**	59.8	35.3	<0.001 ^§^
**Concentric remodeling (%)**	26.2	11.8	
**Concentric hypertrophy (%)**	7.5	11.8	
**Eccentric hypertrophy (%)**	6.5	41.2	
**LA diameter (mm)**	34.6 ± 5.3	43.4 ± 4.1	<0.001 ^#^
**LA area (cm^2^)**	17.7 ± 3.6	25.3 ± 3.8	<0.001 ^#^
**LA volume/BSA (mL/m^2^)**	26.3 ± 7.6	40.9 ± 9.1	<0.001 ^#^
**E/A ratio**	1.22 ± 0.48	1.06 ± 0.40	0.184 ^#^
**e’ septal**	0.09 ± 0.03	0.07 ± 0.01	<0.001 ^#^
**e’ lateral**	0.12 ± 0.04	0.08 ± 0.02	<0.001 ^#^
**Mean E/e’ ratio**	7.22 ± 1.86	11.91 ± 2.07	<0.001 ^#^

^#^ Student *t* test; ^§^ chi squared test; * missing in 1 patient in each group. See Table 1 for abbreviations.

**Table 3 jcm-10-03180-t003:** Logistic regression analysis of factors associated with heart failure with preserved ejection fraction.

		Odds Ratio	95% CI	*p* Value
**PAPS ***	**Univariable**	2.28	0.27–18.86	0.445
**APS-SLE ***		5.77	0.60–55.95	0.131
**PAPS ***	**Sex and age-adjusted**	1.49	0.17–13.44	0.721
**APS-SLE ***		5.93	0.56–63.07	0.140
**Age**	**Univariable**	1.05	1.01–1.09	0.020
**Female sex**	**Univariable**	0.44	0.16–1.24	0.119
**Age**	**Sex and age-adjusted**	1.05	1.01–1.10	0.013
**Female sex**		0.37	0.12–1.08	0.067
**Hypertension**	**Univariable**	25.3	3.3–197.2	0.002
	**Sex and age-adjusted**	20.0	2.4–166.7	0.006
**Diabetes**	**Univariable**	3.22	0.54–19.03	0.197
	**Sex and age-adjusted**	1.81	0.29–11.16	0.522
**Smoking**	**Univariable**	0.82	0.22–3.09	0.768
	**Sex and age-adjusted**	0.92	0.23–3.64	0.903
**Previous arterial events**	**Univariable**	3.36	1.21–9.35	0.020
	**Sex and age-adjusted**	2.67	0.91–7.83	0.074
**Previous VTE**	**Univariable**	0.38	0.14–1.06	0.064
	**Sex and age-adjusted**	0.37	0.13–1.08	0.068
**aCL IgG > 40 GPL**	**Univariable**	3.03	1.04–8.81	0.042
	**Sex and age-adjusted**	3.43	1.09–10.77	0.035
**aCL IgM > 40 MPL**	**Univariable**	1.50	0.44–5.15	0.515
	**Sex and age-adjusted**	1.21	0.33–4.42	0.772
**aβ2GPI IgG > 40 GPL**	**Univariable**	3.10	1.06–9.05	0.038
	**Sex and age-adjusted**	5.28	1.53–18.27	0.009
**aβ2GPI IgM > 40 MPL**	**Univariable**	3.56	1.06–12.01	0.041
	**Sex and age-adjusted**	2.64	0.72–9.67	0.144
**LAC DRVVT > 1.25**	**Univariable**	4.73	1.03–21.77	0.046
	**Sex and age-adjusted**	5.20	1.10–24.68	0.038
**Triple positivity**	**Univariable**	2.56	0.90–7.26	0.078
	**Sex and age-adjusted**	3.56	1.11–11.47	0.033

* Carriers as reference group. APS: antiphospholipid syndrome; aβ2GPI: anti beta-2-glycoprotein-I antibody; aCL: anticardiolipin antibody; DRVVT: dilute Russel’s viper venom time; LAC: lupus anticoagulant; PAPS: primary APS; SLE: systemic lupus erythematosus; VTE: venous thromboembolism.

**Table 4 jcm-10-03180-t004:** Multivariate logistic regression analysis of factors associated with heart failure with preserved ejection fraction.

	Odds Ratio	95% CI	*p* Value
**Age**	1.07	1.00–1.14	0.044
**Female sex**	0.50	0.13–1.85	0.298
**Arterial hypertension**	19.49	2.21–171.94	0.008
**aβ2GPI IgG > 40 GPL**	8.62	1.23–60.44	0.030
**aCL IgG > 40 GPL**	0.65	0.11–3.96	0.640
**LAC DRVVT > 1.25**	2.57	0.43–15.26	0.298

aβ2GPI: anti beta-2-glycoprotein-I antibody; aCL: anticardiolipin antibody; DRVVT: dilute Russel’s viper venom time; LAC: lupus anticoagulant.

## Data Availability

The data presented in this study are available on request from the corresponding author. The data are not publicly available due to privacy reasons.

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
