# Peer review of "Antiphospholipid Antibodies and Heart Failure with Preserved Ejection Fraction. The Multicenter ATHERO-APS Study"

_jcm, 2021, doi:10.3390/jcm10143180_

Round 1
Reviewer 1 Report
This study by Pastori et al looks at the prevalence of HFpEF in patients with APL syndrome. I have enjoyed reading the paper but I feel that the paper misses the main point that would be clinically extremely relevant to healthcare providers.
There is a clear association between hypertension and HFpEF. Can the authors model a multivariate analysis that takes hypertension into account? We need to know if APL is truly associated with HFpEF irrespective of age and hypertension.
If the answer to that critical piece of information is yes, then the highlight of this paper is to demonstrate that APL with aCL IgG, aB2GP1 IgG positivity and LAC DRVTT >1.25 are associated with a higher incidence of HFpEF. Thus those patients need to be monitored more carefully for the development of HFpEF and perhaps the next step is to study the impact of certain therapies such as ACEi/ARB/ARNI or beta-blocker therapy can delay or prevent the occurrence of HFpEF.
Table 3, there is no need to show p values of 95% CI is shown.
Author Response
Reviewer 1
This study by Pastori et al looks at the prevalence of HFpEF in patients with APL syndrome. I have enjoyed reading the paper but I feel that the paper misses the main point that would be clinically extremely relevant to healthcare providers. There is a clear association between hypertension and HFpEF. Can the authors model a multivariate analysis that takes hypertension into account? We need to know if APL is truly associated with HFpEF irrespective of age and hypertension.
Response: Thank for this observation. As suggested, we performed a multivariate analysis with the factors significantly associated with HFpEF, independently from age and sex. By this analysis hypertension and aB2GPI> 40 GPL/ml remained associated with HFpEF (new Table 4). We have addressed the topic in the discussion (line 204-212)
|
|
Odds ratio |
95%CI |
p value |
|
Age |
1.07 |
1.00-1.14 |
0.044 |
|
Female sex |
0.50 |
0.13-1.85 |
0.298 |
|
Arterial hypertension |
19.49 |
2.21-171.94 |
0.008 |
|
aβ2GPI IgG >40 GPL |
8.62 |
1.23-60.44 |
0.030 |
|
aCL IgG >40 GPL |
0.65 |
0.11-3.96 |
0.640 |
|
LAC DRVVT >1.25 |
2.57 |
0.43-15.26 |
0.298 |
If the answer to that critical piece of information is yes, then the highlight of this paper is to demonstrate that APL with aCL IgG, aB2GP1 IgG positivity and LAC DRVTT >1.25 are associated with a higher incidence of HFpEF. Thus, those patients need to be monitored more carefully for the development of HFpEF and perhaps the next step is to study the impact of certain therapies such as ACEi/ARB/ARNI or beta-blocker therapy can delay or prevent the occurrence of HFpEF.
Response: We completely agree with the reviewer: the predictive role of aPL in identifying those APS patients at higher risk of developing HFpEF is the main goal. We added this point to the Discussion section under the paragraph of Implications. Lines 237-239
Table 3, there is no need to show p values of 95% CI is shown.
Response: We think that the presence of p values in the tables could help the readers not too familiar with 95% CI to better recognize the significant variables and the strength of the association. However, if the Reviewer considers this change essential we may amend the manuscript accordingly.

Reviewer 2 Report
This manuscript revealed the relationship HFpEF and APS. These data were interesting. However, I have some questions. Please make clear.
Major point
- The authors write "in addition to typical VTE and arterial thrombosis, HFpEF may be a previously unrecognized clinical presentation of patients with APS". However, this manuscript only revealed that patients may have HFpEF in the case of triple positivity etc. in APS. Please reconsider in discussion and conclusion.
- What is the number of patients with HFpEF in PAPS, APS-SLE and Carriers?
- What are the definitions of HT, DM, VTE and arterial event?
Minor point
- “APS-SLE” and “SLE-APS” are mixed.
- Only NT-pro BNP is used in this manuscript. In Introduction, there is no need to mention BNP.
Author Response
Reviewer 2
This manuscript revealed the relationship HFpEF and APS. These data were interesting. However, I have some questions. Please make clear.
Major point
The authors write "in addition to typical VTE and arterial thrombosis, HFpEF may be a previously unrecognized clinical presentation of patients with APS". However, this manuscript only revealed that patients may have HFpEF in the case of triple positivity etc. in APS. Please reconsider in discussion and conclusion.
Response: We amended our discussion and conclusion according to your suggestion. We highlighted that cardiovascular involvement has been recently discussed as a possible “non criteria manifestation” of APS during the 16th International congress on antiphospholipid antibodies (Sciascia S, Radin M, Cecchi I, Levy RA, Erkan D.Lupus. 2021 Jul;30(8):1314-1326. doi: 10.1177/09612033211020361. Epub 2021 May 27). (references 39 and 40). However, the presence of HFpEF cannot currently still be used as a clinical criteria for APS diagnosis. We changed the sentences regarding the clinical presentation of APS. Please see lines 211-212 and line 251.
What is the number of patients with HFpEF in PAPS, APS-SLE and Carriers?
Response: The proportions are reported in table 1 and are the following: PAPS 12/91 (13.2%); APS-SLE 5/18 (27.8%); Carriers1/16 (6.3%)
What are the definitions of HT, DM, VTE and arterial event?
Response: We added to the method section (line 85-86) the references for the diagnosis of hypertension and diabetes. The type of VTE and arterial events were specified in the results section (line 128-131)
Minor point
“APS-SLE” and “SLE-APS” are mixed.
Response: Thanks, we checked again all the abbreviation.
Only NT-pro BNP is used in this manuscript. In Introduction, there is no need to mention BNP.
Response: We deleted BNP as suggested.

Round 2
Reviewer 2 Report
I have no more comments.